# Multiomics Provide Insights into the Key Molecules and Pathways Involved in the Physiological Adaptation of Atlantic Salmon (*Salmo salar*) to Chemotherapeutic-Induced Oxidative Stress

**DOI:** 10.3390/antiox10121931

**Published:** 2021-11-30

**Authors:** Carlo C. Lazado, Gerrit Timmerhaus, Mette W. Breiland, Karin Pittman, Sigurd Hytterød

**Affiliations:** 1Nofima, Norwegian Institute of Food, Fisheries and Aquaculture Research, 1433 Ås, Norway; gerrit.timmerhaus@nofima.no; 2Nofima, Norwegian Institute of Food, Fisheries and Aquaculture Research, 9019 Tromsø, Norway; Mette.W.Breiland@Nofima.no; 3Department of Biological Sciences, University of Bergen, 5006 Bergen, Norway; Karin.Pittman@uib.no; 4Quantidoc AS, 5006 Bergen, Norway; 5Norwegian Veterinary Institute, P.O. Box 750, Sentrum, 0106 Oslo, Norway; sigurd.hytterod@lakseelver.no

**Keywords:** aquaculture, amoebic gill disease, disinfection, mucosal immunity, oxidative stress

## Abstract

Although chemotherapeutics are used to treat infections in farmed fish, knowledge on how they alter host physiology is limited. Here, we elucidated the physiological consequences of repeated exposure to the potent oxidative chemotherapeutic peracetic acid (PAA) in Atlantic salmon (*Salmo salar*) smolts. Fish were exposed to the oxidant for 15 (short exposure) or 30 (long exposure) minutes every 15 days over 45 days. Unexposed fish served as the control. Thereafter, the ability of the remaining fish to handle a secondary stressor was investigated. Periodic chemotherapeutic exposure did not affect production performance, though survival was lower in the PAA-treated groups than in the control. Increased ventilation, erratic swimming, and a loss of balance were common behavioural manifestations during the oxidant exposure. The plasma reactive oxygen species levels increased in the PAA-treated groups, particularly after the third exposure, suggesting an alteration in the systemic oxidative stress status. Plasma indicators for internal organ health were affected to a certain degree, with the changes mainly observed after the second and third exposures. Metabolomics disclosed that the oxidant altered several circulating metabolites. Inosine and guanosine were the two metabolites significantly affected by the oxidative stressor, regardless of exposure time. A microarray analysis revealed that the gills and liver were more responsive to the oxidant than the skin, with the gills being the most sensitive. Moreover, the magnitude of the transcriptomic modifications depended on the exposure duration. A functional analysis showed that genes involved in immunity and ribosomal functions were significantly affected in the gills. In contrast, genes crucial for the oxidation-reduction process were mainly targeted in the liver. Skin mucus proteomics uncovered that the changes in the mucosal proteome were dependent on exposure duration and that the oxidant interfered with ribosome-related processes. Mucosal mapping revealed gill mucous cell hypertrophy after the second and third exposures, although the skin morphological parameters remained unaltered. Lastly, repeated oxidant exposures did not impede the ability of the fish to mount a response to a secondary stressor. This study provides insights into how a chemical oxidative stressor alters salmon physiology at both the systemic and mucosal levels. This knowledge will be pivotal in developing an evidence-driven approach to the use of oxidative therapeutics in fish, with some of the molecules and pathways identified as potential biomarkers and targets for assessing the physiological cost of these treatments.

## 1. Introduction

The continuous growth of fish farming is perennially challenged by diseases that present a significant impediment to the profitability and sustainability of the industry. Preventive measures to improve and safeguard health have been focal strategies in recent years, including heightened biosecurity [1], the use of balanced and functional diets [2], vaccines [3], and efficiently controlling the rearing environment [4]. However, there are instances where the use of therapeutics remains the only viable option. A significant portion of the chemotherapeutics being used in aquaculture target bacterial and parasitic infections, and for many of these, effective vaccines are not yet available. Though chemotherapeutics can address the health issue in several ways, there have been long-standing discussions on whether the application of these compounds fosters a sustainable industry [5,6], especially given that resistance, imprudent usage and discharge, and unwanted ecological impacts are daunting challenges [7,8]. Mitigating these risks will include implementing stricter rules, streamlined application backed up by research data, and the continuous exploration and development of more eco-friendly alternative therapeutics [9,10].

Oxidative biocides are one of the widely known groups of chemical therapeutics being used in fish farming. In particular, hydrogen peroxide (H_2_O_2_) and peracetic acid (PAA) are considered “green” therapeutics because they have been shown not to contribute harmful residues to the environment [9]. Unlike H_2_O_2,_ which is readily available in a pure form, commercial PAA is an equilibrium mixture of PAA, H_2_O_2_, acetic acid, and water [10]. Both share a similar mechanism of action as biocides, which is the generation of free hydroxyl radicals that induce damage to DNA, enzymes, and proteins via oxidation, thus increasing the permeability of the cell walls [11]. Cell wall destruction may involve different targets, including the peroxidation and disruption of membrane layers, oxidation of oxygen scavengers and thiol groups, inhibition of enzymatic activity, oxidation of nucleosides, impaired energy production, disruption of protein synthesis, and eventually, cell death [12]. These mechanisms contribute to the strong biocidal properties of H_2_O_2_ and PAA, which have been identified to have broad-spectrum activity against aquaculture-relevant pathogens [11,13,14]. However, because of its fat solubility, PAA has far more potent antimicrobial properties than H_2_O_2_ [15]. In addition, concerns have been raised regarding the excessive use of H_2_O_2_, as in Atlantic salmon (*Salmo salar*) farming, due to toxicity threats to other organisms, particularly shrimp [16]. PAA degrades relatively faster than H_2_O_2_, and the difference in the degradation kinetics is important for the PAA by-products not to persist for an extended period in the environment and present risks to other organisms [10]. Hence, PAA offers some promising advantages over H_2_O_2_. Concentrations from 0.2 to 14 ppm have been evaluated in different fish species and responses to the PAA biocide are dictated by dose, application method and duration, stress status of fish and water chemistry [14,17,18].

The use of these oxidative biocides must account for the effects on the environment and target pathogens as well as the fish. It is crucial that the therapeutic doses being applied are effective against the target pathogen and, most importantly, present little to negligible effects on the host fish. This balance is often difficult to achieve because of our limited understanding of the physiological aspects of oxidative biocide application in fish. Nonetheless, several reports in recent years have offered insights into the mechanisms by which PAA interferes with the different physiological processes of salmonids. Biological data have raised several questions regarding the extent of its influence on fish health and welfare [8,13,17,18,19,20]. Transient and chronic oxidative stress has been shown to be induced following PAA treatment, where the enormity of impact is dependent on dose and exposure duration. Therefore, despite its biocidal effectivity, PAA can also be regarded as a potential oxidative stressor. Understanding the magnitude of and how PAA induces oxidative stress, particularly the underlying processes and mechanisms, will be vital for its evidence-driven use in fish farming.

Here we report the physiological consequences of repeated exposures to the oxidative biocide PAA in Atlantic salmon (*Salmo salar*) smolts. Most treatment studies in fish have dealt with a single exposure to the therapeutics, but in a real-world scenario, fish are treated several times during a production cycle. The simulation performed in this study mimicked a treatment for a gill ectoparasite in salmon using a therapeutic being explored as a treatment option [18]. In this trial, we addressed the impacts on uninfected naïve salmon to profile the baseline physiologic response to repeated therapeutic interventions.

## 2. Materials and Methods

### 2.1. Ethical Use of Animals for Research

All procedures involving fish described in this paper followed the Directive 2010/63/EU as amended by Regulation (EU) 2019/1010. The trial received approval from the Norwegian Food Safety Authority under FOTS ID 19321. Key personnel in the trial have FELASA C certification.

### 2.2. Oxidative Biocide Exposure Trial and Secondary Stress Test

The fish trial was conducted at the Tromsø Aquaculture Station (HiT; Tromsø, Norway) using Atlantic salmon smolts produced at the research station. Three hundred and sixty smolts with a starting weight of approximately 80–90 g were stocked into nine 500 L circular tanks in a flow-through system at a density of 40 fish per tank (Figure 1). Fish were allowed to acclimatise for a week under the following parameters, which were also maintained throughout the trial: water flow rate set at 6–7 L·min^−1^, salinity at 35‰, temperature at 13.0 ± 1 °C, dissolved oxygen > 90% saturation, photoperiod set at 24 L: 0 D, and a continuous feeding regime (Nutra Olympic 3 mm, Skretting, Averøy, Norway). There were three experimental groups—control, short exposure (SE), and long exposure (LE). Three tanks were dedicated for each treatment, which were randomly allocated inside the experimental hall.

Oxidant exposure was performed as follows. Water flow in the tank was stopped, and the oxidative biocide (Divosan Forte™, Lilleborg AS, Oslo, Norway) was added to the water column to achieve a concentration of 10 mg L^−1^. This concentration is twice the dose previously used for salmon [18]. Aeration was supplied to allow mixing and maintain oxygen levels > 90%. For the SE group, the exposure duration was 15 min, while for the LE group, the exposure lasted 30 min. After the exposure period, the water flow was opened, and at least 90% of the water was replaced within 8–10 min to flush out the residuals. The control group was unexposed. This exposure protocol was repeated every 15 days, with 3 exposures in total over a 45-day trial period.

A week after the last sampling, the remaining fish were subjected to acute stress by lowering the water volume in the tank to achieve a density 10× higher than the initial density. During the exposure period, the oxygen level in the tank was routinely followed and maintained at above 90% saturation. After exposing the fish to this condition for 1 h, the water level was returned to the initial level, and the fish were allowed to recover for post-stress sampling.

### 2.3. Sampling Protocols

For the main exposure experiment, sampling was performed 24 h after each exposure. Briefly, 3 fish were taken from each tank and humanely euthanised with an overdose of Benzocaine (Benzoak vet, 200 mg/mL, EuroPharma, Leknes, Norway). The length and weight were measured, and the external welfare status was assessed [21]. Plasma samples were collected from blood drawn from the caudal artery using a heparinised vacutainer (BD Vacutainer™, Loughborough, UK). Skin mucus samples from both sides of the fish below the lateral line were collected using FLOQSwab^®^ (COPAN Diagnostics, Murrieta, CA, USA) and snap-frozen in dry ice. Sections of the skin below the dorsal fin and the second gill arch were collected for microarray and mucosal mapping analyses. Samples for the microarray were suspended in RNAlater^®^ (Merck Life Science AS, Oslo, Norway) and stored at −70 °C until analysis, whereas tissues for mucosal mapping were preserved in neutral buffered formalin (BiopSafe ApS, Hellerup, Denmark). A portion of the liver was also collected and stored in RNAlater^®^. For the post-exposure stress trial, plasma samples were taken from 3 fish per tank before the stress and 2 and 4 h after the stress following the protocol described above.

### 2.4. Plasma Clinical Biochemistry

The lactate, glucose, aspartate aminotransferase (ASAT), alanine aminotransferase (ALAT), and creatinine plasma levels were measured with a Pentra C400 Clinical Chemistry Analyzer (HORIBA ABX SAS, Montpellier, France), while the cortisol (Demeditic Diagnostics GmbH, Kiel, Germany) and reactive oxygen species (ROS)/reactive nitrogen species (CellBiolabs, Inc., San Diego, CA, USA) were measured using commercially available kits. All analyses were run in duplicate.

### 2.5. Plasma Metabolomics

Proteins in the plasma (ca. 200 µL) were initially precipitated using methanol followed by a chloroform and water liquid–liquid extraction and collection of the aqueous phase. The extracts were transferred to a liquid chromatography (LC) vial and dried under nitrogen flow. The LC/mass spectroscopy (MS) analysis was performed by MS Omics ApS (Vedbæk, Denmark) in an ultra-performance LC system (Vanquish, Thermo Fisher Scientific, Waltham, MA, USA) coupled with a high-resolution quadrupole-orbitrap mass spectrometer (Q Exactive™ HF Hybrid Quadrupole-Orbitrap, Thermo Fisher Scientific) using a slightly modified version of an earlier protocol [22] used for salmon plasma [8,18]. Data were processed using Compound Discoverer 3.0 (Thermo Fisher Scientific), and the metabolites were identified with four levels of annotation as described in detail in an earlier publication [18].

### 2.6. Microarray Analysis—Mucosal Organs and Liver

Total RNA was isolated from the RNAlater^®^-preserved samples using the Agencourt RNAdvance™ Tissue Total RNA Purification Kit (Beckman Coulter Inc., Brea, CA, USA). All samples had an RNA Integrity Number (RIN) above 9.0 as evaluated by the Agilent^®^ 2100 Bioanalyzer™ RNA 6000 Nano Kit (Agilent Technology Inc., Santa Clara, CA, USA). The microarray analysis was performed using a custom-designed 15K Atlantic salmon DNA oligonucleotide microarray SIQ-6 (Agilent Array, ICSASG_v2), and all reagents used were from Agilent Technologies. The One-Color Quick Amp Labelling Kit was used for RNA amplification and Cy3 labelling using 110 ng of RNA template per reaction. Gene Expression Hybridization Kits were used for the fragmentation of labelled RNA. This was followed by a 15 h hybridisation in a 65 °C oven with a constant rotational speed of 10 rpm. Thereafter, the arrays were successively washed with Gene Expression Wash Buffers 1 and 2 and scanned using the Agilent SureScan Microarray Scanner. Pre-processing was performed in Nofima’s bioinformatics package STARS (Salmon and Trout Annotated Reference Sequences) [23].

### 2.7. Skin Mucus Proteomics

Skin mucus peptides were prepared using a double-digestion protocol (Pierce™ Mass Spec, Thermo Fischer, USA) that was slightly modified for mucus samples. Skin mucus lysates were reduced for 45 min in 10 mM dithiothreitol at 50 °C, alkylated for 20 min in 50 mM iodoacetamide at room temperature, and the proteins were acetone-precipitated overnight at 4 °C. The protein pellet was then digested for 2 h in 1 µg Lys-C endoproteinase at 37 °C followed by a trypsin treatment at 37 °C overnight. The samples were frozen at −80 °C to stop digestion and concentrated in a vacuum evaporator. The protein digests were subjected to an LC-MS analysis, and the resulting spectral data were analysed using Mascot (Matrix Science, London, UK; version 2.6.1). The MS/MS-based peptide and protein identifications were validated in Scaffold (version Scaffold_4.8.9, Proteome Software Inc., Portland, OR, USA) following the pipeline detailed in an earlier publication [24].

### 2.8. Mucosal Mapping

The gill and skin samples were processed for Mucosal Mapping following Quantidoc’s standard protocol [25]. Tissue sections stained by Periodic Acid Schiff—Alcian Blue were digitised, scanned, and processed through an automated software developed by Quantidoc AS for the stereological image analysis of mucosa. The analysed mucosal features include mucous cell density (D, % epithelium filled with mucous cells), mean mucous cell area (A, μm^2^), and barrier status (1/[A:D] × 1000) [25,26,27,28].

### 2.9. Data Handling and Statistics

Statistics were performed in Sigmaplot 14.0 Statistical Software (Systat Software Inc., London, UK). A Shapiro-Wilk test was used to evaluate the normal distribution, and a Brown-Forsyth test was used to check for equal variance in the data set. Plasma parameters were subjected to a two-way analysis of variance (ANOVA) followed by multiple pairwise comparisons with the Holm-Sidak test to identify differences between treatment groups at a particular time point, differences within a treatment group over time, and the interaction of treatment and time. All tests for statistical significance were set at *p* < 0.05.

For the microarray, data normalised by the Lowess normalisation of Log2-expression ratios in STARS [23] were subjected to statistical comparisons using linear modelling as implemented in the Bioconductor package limma [29]. Significance values (*p* values) were adjusted for multiple testing using the Benjamini-Hochberg procedure. The differential gene expression significance cut-off was an adjusted *p*-value < 0.01. For each comparison, positive and negative fold changes indicated the up- and down-regulation, respectively, of gene expression in the oxidant-exposed group relative to the control group. A hypergeometric test was used to identify the gene ontology (GO) terms in which significant genes (defined using the adjusted *p*-value < 0.01) were over-represented. This was performed for all three ontologies: biological process, molecular function, and cellular compartment. Zebrafish orthologs for the Atlantic salmon genes were retrieved from the Ensembl Compara database (https://www.ensembl.org/info/genome/compara/, accessed on 20 May 2020). The zebrafish orthologs of the significant genes from each comparison were then analysed for enrichment in the Reactome pathways (www.reactome.org, accessed on 20 May 2020) using a hypergeometric test. Each GO term or Reactome pathway enrichment (*p* < 0.05) was assessed for up- and down-regulated genes separately, as well as for the union of up- and down-regulated genes.

For the plasma metabolomics data, a multivariate principal component analysis (PCA) model was used to identify the effects influenced by different treatments. A Benjamini–Hochberg correction was then employed with the acceptable false-positive rate set at 0.1. The Benjamini–Hochberg critical value, (i/m)Q, where i = the individual *p*-value rank, m = total number of tests, and Q = the false discovery rate, was calculated for each compound. A compound was considered significantly affected by the treatment when the *p*-value from a *t*-test was <(i/m)Q. Furthermore, a Log2 ratio was calculated by comparing the level in the PAA-treated group with the control.

For the proteomics analysis, data were processed and analysed in R (version 3.5.2, https://www.r-project.org/, accessed on 20 May 2020). Low abundant results were removed by deleting all proteins with mean counts smaller than or equal to 1. The remaining proteins were normalised by dividing the counts by the individual mean counts. To identify differences between groups, ANOVAs were calculated for each protein. A *p*-value of <0.05 was used as the cut-off for filtering. These proteins were prepared for a cluster analysis by calculating the group means, centred by dividing by row means, and Log2 transformations. The data were clustered by the function *hclust() (stats* package, for Euclidean distance and with complete linkage) and plotted with *heatmap.2() (gplots* package). Five sub-clusters with distinctive expression patterns were identified, and the mean values of these were plotted as bar plots with error bars showing the standard error of the mean (SEM).

## 3. Results

### 3.1. Behaviour, Production Performance, External Welfare, and Survival

There were three prominent behavioural changes observed during the exposure period: erratic swimming, increased opercular ventilation, and a loss of balance. These changes could be arbitrarily divided into three periods: first 10 min—active and erratic swimming activity with some fish attempting to jump out of the water; the following 10 min—slightly diminished swimming activity, some fish attempting to burrow at the base or side of the tank, and at least 60% of the fish demonstrating increased opercular ventilation; and the last 10 min—rapid opercular ventilation with at most, 10% of the fish exhibiting a loss of balance. These behavioural changes were documented for all exposure events.

Production parameters, including length and weight at termination, did not significantly change among the treatment groups, although there were some stochastic changes in different welfare indicators after the 3rd exposure (Figure 1B). Nonetheless, the overall external welfare status of the experimental fish remained favourable, as the average score was below 1 (Figure 1C). We also observed that the fish resumed feeding immediately after treatment for all exposure events.

There was one dead fish 24 h after the 1st and 2nd exposures in the LE group. A total of 6 dead fish were recorded in the SE group, while 8 were recorded in the LE group before conducting the 3rd exposure. After the 3rd exposure, 6 and 5 dead fish were recorded in the SE and LE groups, respectively. The survival rates of the oxidant-exposed fish were relatively lower than that of the control group at the termination of the experiment.

### 3.2. Plasma ROS Levels

The plasma ROS levels significantly increased through time in the SE and LE groups, but not in the control group (Figure 1D). However, it was only after the 3rd exposure that the ROS levels in the plasma of the oxidant-exposed groups significantly differed from the control. In addition, no significant difference was observed between the two oxidant-exposed groups.

### 3.3. Indicators of Systemic Stress and Organ Health

Of the three physiological stress indicators, namely cortisol (Figure 2A), lactate (Figure 2B), and glucose (Figure 2C), only glucose showed significant changes following exposure to the oxidant. The glucose level in the LE group was significantly lower than those in the SE and control groups after the 2nd exposure. However, this change was not identified after the 1st and 3rd exposures. Moreover, the glucose level in the LE group was significantly lower after the 2nd exposure than after the 1st and the 3rd exposures.

The ASAT levels (for liver health) showed significant temporal changes in all experimental groups (Figure 2D), with the level in the SE group significantly higher than that in the control after the 1st and 3rd exposures. However, the ASAT levels in both oxidant-exposed groups was significantly lower than that in the control group after the 2nd exposure. Alanine aminotransferase, another liver health indicator, significantly changed after the 3rd exposure, where an increase was identified in the SE group relative to the control (Figure 2E). Both oxidant-treated groups had higher ALAT levels after the 3rd exposure than after the 1st exposure. Creatinine (for renal health) significantly increased over time in all groups (Figure 2F), with a 40-fold increase between the 1st and 2nd exposures, particularly in the control group. There was at least a 13-fold increase in the creatinine level in all groups after the 3rd exposure compared with the 1st exposure. A significant inter-treatment difference was also identified after the 2nd exposure, where the levels in the SE and LE groups were lower than that in the control.

Subjecting the oxidant-exposed fish as well as the control group to a secondary stressor (i.e., crowding stress) resulted in a significant increase in plasma cortisol post-stress (Figure 2G), where in terms of increment, SE group showed the most pronounced response after stress. The plasma cortisol levels in the SE group were significantly higher those in the control and LE groups at 2 and 4 h post stress. The lactate (Figure 2H) and glucose (Figure 2I) levels remained unchanged following the crowding stress.

### 3.4. Alterations in the Circulating Metabolites

A total of 944 compounds were detected in the plasma samples collected after the 3rd exposure, of which 197 were annotated at Level 3, 75 at Level 2b, 23 at Level 2a, and 38 at Level 1 (Appendix A). The score plot from a PCA model calculated on the compounds annotated at Level 1, 2a, or 2b in the reduced dataset is presented in Figure 3A, and the loading plot (Figure 3B) shows which variables are responsible for the patterns observed in the score plot. The grouping of the samples shows a clustering of the two oxidant-exposed groups in the upper part, separated from the control samples, which are all located in the lower part of the plot. Comparing the SE group with the control group, there were six differentially affected metabolites: inosine, 7-methyladenine, biotinsulfoxide, 4-acetamidobenzoic acid, hypoxanthine, and guanosine. Similarly, 5 differentially affected metabolites were identified in the LE group, namely 4-acetamidobenzoic acid, inosine, valpromide, 7-methyladenine, and guanosine. Inosine (Figure 3C, Level 1) and guanosine (Figure 3D, Level 2a), two metabolites with higher annotation confidence, were commonly affected in the oxidant-exposed groups, with an up-regulation observed in the former and a down-regulation identified in the latter.

### 3.5. Oxidant-Induced Transcriptomic Changes in the Gills, Skin, and Liver

The number of differentially expressed genes (DEGs) identified in the SE group was comparable across all three tissues (15 DEGs in gills, 11 in skin, and 23 in liver; Figure 4A). Notably, none of these DEGs was significantly changed in more than one tissue after 15 min of exposure (Figure 4B). Increasing the duration of oxidant exposure from 15 to 30 min led to a marked increase in the numbers of DEGs identified in the gills (i.e., 693 DEGs). Twelve DEGs significantly overlapped in the gills of the SE and LE groups, including *mucin-7-like*, *IL15 receptor alpha chain isoform 3,* and *C-C motif chemokine 4-like* (Figure 4B). The increase in DEGs in the LE group was also true for the liver, but to a lesser extent, with the number of DEGs increasing from 23 to 100. There were 13 hepatic DEGs that significantly overlapped between the SE and LE groups. Of the three tissues, the skin was least responsive to the oxidant exposure in terms of the number of DEGs identified, with only 13 DEGs identified in the LE group. The transcriptomic changes in the gills and liver were further reflected in the magnitude of responses as shown in volcano plots (Figure 4C–F). The majority of the DEGs in the gills of the SE group were up-regulated (Figure 4C), while a pronounced down-regulation profile was identified in the liver (Figure 4E). For the LE group, the distribution of up- and down-regulated genes was almost equal in both the liver and gills (Figure 4D,F).

The functional annotations by Gene Ontology (Figure 5) and Reactome Pathway (Figure 6) were focused on the gills and liver. The analysis for the skin yielded no significant results due to the low number of DEGs. No significantly enriched GO terms were identified in the gills of the SE group. In the gills of the LE group, however, the DEGs identified were enriched in pathways related to immunity such as “chemokine activity” and “immune response”, and pathways related to ribosomal function such as “ribosome biogenesis”, “rRNA processing”, and “RNA binding” (Figure 5B). Enriched pathways in the livers of the SE group included “nucleotide binding” and “ATP binding” (Figure 5A). Extending the exposure duration, the DEGs in the livers of the LE group were enriched in pathways such as “oxidoreductase activity” and “oxidation-reduction process” (Figure 5C).

The Reactome analysis revealed that the zebrafish orthologs of the DEGs identified in the gills of the LE group were enriched in pathways related to the immune system and the ribosome (Figure 6A,B). This mirrored the findings of the GO term enrichment (Figure 5). In detail, the pathways enriched in zebrafish orthologs included “rRNA processing”, “mitochondrial translation”, “interleukin-20 family signalling”, and “deubiquitination”.

### 3.6. Alterations in the Skin Mucus Proteome

There were 1333 proteins identified in the samples, and downstream analyses focused on the 189 differentially expressed proteins. A cluster analysis based on how the skin mucus proteins were affected by the oxidant revealed five major clusters (Figure 7A). Cluster 1 includes proteins that were significantly up-regulated in the SE group but down-regulated in the LE group. Clusters 2 and 4 include proteins that were differentially up-regulated by oxidant exposure. In Cluster 2, the effects were more substantial in the SE group than in the LE group. Cluster 3 includes proteins that were significantly down-regulated following oxidant exposure, with the magnitude of regulation higher in the LE group than in the SE group.

There were 69 differentially expressed proteins exclusively found in the LE group and 46 in the SE group (Figure 7B). The LE and SE groups shared 14 differentially expressed proteins. We further investigated these to identify overlaps according to the direction of change. There were a relatively higher number of shared up-regulated than down-regulated proteins between the two oxidant-exposed groups (Figure 7C). We performed a STRING analysis to understand the protein-protein interactions and functional groupings of the proteins (Figure 7D). The protein interaction network created 115 nodes and 143 edges, with an average node degree of 2.49. We found that proteasome (yellow), ribosome (red), and spliceosome (brown) KEGG pathways were significantly enriched. Looking into the interactions, the network highlights the interaction of ribosomal proteins (rpl38, eif3m, rps23, rpl30, rpl14, rpl39, rps14, rpl26, rpl10a, rps11, rpl35a, rpl27, rpl36), ubiquitins (psmc2, psmd8, psmd3, psmc4, psmb2, psmc1b), proteasomal proteins (psmd10, psmc2, psmd8, psmd3, psmc4, psmb2, psmc1b), fibrin clot formation (fgb, fgg), growth factors (fgb, f2, fgg, zgc:161979, zgc:113828), and proteins with a role in ROS detoxification (zgc:56493, prdx5).

### 3.7. Mucous Cell Morphometry

Mucous cell morphometries, including area, volumetric density, and barrier status, were measured in the gill lamellae after each exposure (Appendix A), but only after the 3rd exposure in the dorsal skin (Appendix A). There was a significant temporal change in the mucous cell area in the gill lamellae. The highest measurement was recorded after the 2nd exposure (Appendix A), though no significant inter-treatment differences were found. After the 3rd exposure, the mucous cells were significantly larger in the LE group compared to the control and SE groups. For the other two parameters, no significant temporal or inter-treatment differences were identified in the gill lamellae (Appendix A). None of the measured mucous cell parameters were changed in the dorsal skin after the 3rd exposure (Appendix A).

## 4. Discussion

When not managed properly, chemical oxidative stressors (often via chemotherapeutics) can be detrimental to fish health and welfare. Therefore, a holistic understanding of how fish respond to these stressors is important to developing an evidence-driven approach for their application in fish farming. This paper explored the molecules and processes involved in how Atlantic salmon respond and adapt to chemical therapeutics. Through a multi-platform approach, we demonstrated that repeated exposures at longer durations impact both the mucosal and systemic responses in salmon, with the gills identified as highly susceptible to chemotherapeutics.

The behavioural changes observed during the exposure trial revealed that fish recognise danger and exhibit avoidance activities to escape contact with the oxidant, which is consistent with earlier observations [8]. Prolonging the contact time with PAA highlighted some of the classic behavioural changes when fish encounter chemical stimuli [30], including strong respiratory pressure as clearly demonstrated by opercular hyperventilation and eventually a loss of equilibrium. These changes indicate some of the welfare risks of using this chemical oxidant in salmon, including the mortality observed after the treatments. However, the oxidant-exposed fish recovered quickly, and the overall production performance and external welfare index were not significantly affected after 3 exposure events, suggesting that though it posed a mild risk, application optimisation could perhaps mitigate untoward consequences. In addition, the classical plasma stress indicators after each exposure were not significantly affected, implying the fish either adapted quickly or the treatment did not incite a strong stress response, which is slightly contradictory to our earlier observations using a lower PAA dose [20]. The lower glucose level in the LE group after the 2nd exposure could indicate interference in energy metabolism; however, this was likely a transitory response as the profile was not consistent throughout the trial.

We have previously shown that the oxidant used in this experiment (i.e., PAA) could trigger transient oxidative stress in salmonids, as shown either by the activation of antioxidant defences or by elevated or dysregulated levels of internal ROS [8,13,19,20]. This response is predominantly attributed to the dissociation of the by-products of PAA when it comes into contact with water, which includes the generation of free hydroxyl radicals [10,31]. The production of free radicals makes PAA a therapeutic with potent and broad microbicidal functions [32]. We found that exposure to the oxidant triggered the endogenous generation of ROS, with both treated groups exhibiting elevated levels, but only after the 3rd exposure. At present, we cannot conclusively account for whether the exogenous ROS from PAA degradation directly contributes to this increase. The increase in internal ROS indicates that the biocide triggered oxidative stress and points to the possibility that this consequence was likely cumulative, because the change relative to the control was pronounced only after three exposure events. Nonetheless, signs of the increasing tendency were already evident after the 2nd exposure. The neurobiology behind salmon’s responses to PAA is an exciting avenue for future studies given the fact that the classical stress indicators of the hypothalamic-pituitary-adrenal axis (Figure 2) and the oxidative stress marker (Figure 1) did not show an apparent agreement. The interaction of these systems in fish is not yet elucidated, but in mammals, a link has been demonstrated [33]. Nevertheless, the plasma ROS profiles after the first two exposure events suggest the fish may be able to effectively regulate internal ROS homeostasis following treatment with an oxidative biocide, though such an ability was compromised after successive exposures. This internal ROS homeostasis regulation is likely mediated through the activation of molecules that participate in antioxidant mobilisation and/or excess ROS scavenging, which are discussed in detail in the next sections.

When an organism is under stress, several physiological adaptations are initiated to counteract the threats, including modifications to the thousands of metabolites necessary for homeostasis and adaptation [34]. These metabolites engage in different roles such as directly interacting with the stressors (e.g., ROS) by neutralising them, thereby making them less detrimental to the organism and/or they will ensure that the different physiological systems are able to withstand the pressure of the stressful episode (e.g., immunity, energy metabolism). The systemic impact of the chemical oxidant was further clarified by the metabolomic changes in the plasma. Oxidant exposure altered the salmon’s plasma metabolome, though these changes were not dependent on exposure duration. These metabolomic modifications may be an essential systemic buffering mechanism responding to the physiological dysregulation from a ROS level imbalance [35]. Compared with earlier studies in which a lower PAA dose was employed (Lazado et al., 2020; Lazado et al., 2021), the number of altered metabolites was relatively lower, indicating either the impact on the salmon metabolome may not be aggravated by a higher dose or that the treatment desensitised the ability of salmon to mount countermeasures to the physiological pressures from PAA.

Inosine and guanosine were the two metabolites identified to be significantly affected in both the SE and LE groups, suggesting their key role in salmon’s adaptive response to the chemical oxidant. We earlier identified that plasma guanosine is altered when salmon are exposed to a similar oxidant but at a lower dose [18], thus underlining the fundamental function of this molecule in response to chemically induced oxidative stress in this species. These molecules have known distinct roles in mitigating oxidative damage caused by ROS by protecting cells from DNA damage [36]. In particular, in fish, dietary inosine may confer oxidative stress resistance [37]. Therefore, the regulation of these two metabolites likely played a role in ameliorating the effects of the imbalanced internal ROS that resulted in oxidative stress, likely in a similar mechanism described in higher animal models. To explore the potential of these metabolites as biomarkers for stress in salmon, it is important to identify the baseline levels. The metabolomic profile for fish is often highly influenced by the handling associated with sampling [38] and, hence, a standard framework for salmon plasma metabolite collection and interpretation would be beneficial.

The gills are far more sensitive to the oxidant than the skin, as demonstrated by both transcriptomics and, to some extent, by mucosal mapping, where the changes in the gills provided a clear exposure duration-dependent response profile. This study supports the striking difference between the responses of salmon gills and skin to a chemical oxidant. Previous studies have indicated that oxidative agents induce more pronounced morphomolecular changes in salmon gills relative to the skin, and this difference has been attributed to the inherent large branchial surface area exposed to the environment and the greater structural complexity of the gills [18,25,28,39]. Mucosal surfaces such as the gill and skin mucosa provide the first line of defence by providing both structural protection and an array of molecules vital for defensive and adaptive responses under challenging conditions. One of its cellular components, the mucous cells, produce the slimy mucous layer and exhibit phenotypic plasticity under stress [25,40]. Though not as evident as observed in earlier studies [25,28], the mucous cells in the gills increased in size, especially in the LE group, following exposure to the oxidative stressor, which indicates remodelling that may be crucial for mucosal barrier integrity. However, it is possible that the marginal response of the mucous cells points to impeded normal physiological function under the stressful conditions induced by the periodic application of the chemotherapeutics.

There is a tight relationship between immunity and oxidative stress in fish, particularly in ensuring that ROS remain below the detrimental level. Antioxidants are mobilised for neutralisation and scavenging, and other defence factors are activated to provide the fish with protection from oxidative stress-related damage [13,19,20,40]. The functional annotations through the GO and Reactome analyses both highlighted that the oxidative chemotherapeutics impacted gill mucosal immunity, especially molecules involved in immune cell-cell communications. Some of the DEGs that significantly overlapped between the SE and LE groups, notably in the gills, have key functions in mucosal defence, including *il15r**α*, the putative receptor for IL-15. The roles of IL-15L and IL-15Rα in Type 2 immunity have recently been reported in fish [41], and this study points to the possibility that it may also be important in immunity to oxidative stress. In mammalian systems, the IL-15 myokine has been implicated to have a protective effect against H_2_O_2_-mediated oxidative stress [42]. Mucins are the major glycopolymeric components of mucus and represent a large class of proteins in vertebrates, and *Muc7* was found to be commonly regulated in both the SE and LE groups. It is a secreted non-oligomerising mucin that can self-aggregate, but it is not thought to contribute to the mucus properties in mammalian models [43]. Its functional role remains elusive in fish, but its strong regulation in the present study provides a potential connection to its role during chemically induced stress.

The ribosome is a ribonucleoprotein-based molecular machine that orchestrates protein biosynthesis, and this complex process presents numerous control points for stress response regulation [44]. We found that pathways related to ribosomes were significantly enriched in the gills of the LE group. Ribosomal proteins were similarly overly represented in the skin mucus proteome, where a pronounced hypothetical protein-protein interaction was observed. The significant changes in the number of affected ribosomal genes and proteins, including the magnitude of their alterations, demonstrate their role in the adaptive cellular responses to the oxidative chemotherapeutics, which may impact ribosome dynamics and function. Though the interplay of oxidative stress and ribosome function has not been fully elucidated in fish, it has been shown in other organisms that oxidative stress affects protein translation such as during translational errors [44]. Ribosomal proteins repair the damage, thereby ensuring ribosome homeostasis and stability as the organism adapts to the condition. We believe that such a mechanism may be at play in the gill and skin mucosa to counteract the pressure from the chemotherapeutics. The effects of the oxidant on the skin mucosa were clearer in the skin mucus proteome than in the skin tissue transcriptome. Though the difference in the platforms and bioinformatics strategies used could explain this disparity, it is also likely that post-translational modifications may have a significant impact on skin mucosal responses to PAA, which is an area of interest for future studies.

The liver has a major function in the detoxification and maintenance of the body’s metabolic homeostasis. We found that repeated exposures to the chemical oxidant presented a risk to liver function, as elucidated by both the clinical biochemistry and transcriptomics. The ASAT and ALAT levels were affected by the chemical oxidant, with the changes more pronounced in the SE group, suggesting that the interference in liver function was likely not dose dependent. These plasma indicators are used as tools to detect liver disturbances in salmonids, and increased levels may indicate impaired liver function, liver tissue damage, and necrosis [45]. The levels are considerably influenced by pre-analytical causes for variation such as diet, stress, and production site, which could become a confounding factor in the interpretation of the data. The values documented in this study were lower than the proposed baseline levels for adult salmon (>3.5 kg) [46]. The specific clinical significance of these analytes in salmon health monitoring is yet to be fully substantiated, as clinical biochemistry is an approach that is not widespread in aquaculture. We hope that the values identified here will be beneficial to establishing the standards and biological relevance of these analytes in fish. The hepatic transcriptome provides indication that the redox balance in the liver was altered by the chemical oxidant, as pathways related to oxidoreductase activity and the oxidation-reduction process had been altered. This further indicates that repeated exposures to the chemical oxidant not only trigger mucosal oxidative stress but may also be impacting internal organ oxidative status and a predisposing factor for the increased ROS levels in the plasma. There was also an indication that the oxidant exposure may influence renal function, with lower plasma creatinine levels after the 2nd exposure. Nonetheless, it is difficult to draw an implication because the control group also had a substantially higher creatinine level, and the effects were only observed at one time-point. We checked all parameters before the 2nd exposure that could explain the elevated creatinine in the control group but could not pinpoint a significant deviation.

We had previously found that crowding stress prior to treatment modifies the responses of salmon to chemotherapeutics [18]. Here, we tested whether exposure to the oxidant could alter the responses of salmon to a secondary stress such as crowding, which is a common stressor in fish farming. Repeated exposures to the oxidant did not significantly impact the salmon’s responses to a secondary stressor, as they were able to regulate plasma cortisol levels similarly to the unexposed fish. However, it was interesting to observe that the SE group had higher cortisol levels than the two other groups, indicating that short-term exposure to the oxidant may influence the kinetics after a stressful episode. Overall, the post-stress responses suggest that repeated exposure did not pose a considerable risk to the ability of the salmon to respond to a secondary stressor.

## 5. Conclusions

This study demonstrated that periodic exposures to oxidative chemotherapeutics affect salmon physiology at different magnitudes. The chemical oxidant altered the balance of internal ROS and consequently, triggered systemic oxidative stress. Localised oxidative stress was likewise induced in the mucosal organs, particularly in the gills. A similar impact was also displayed in the hepatic transcriptome. Gills were found to be more highly sensitive to the chemical oxidant than the skin and may be the organ that plays a crucial role in the adaptive mucosal response to a chemical stressor. Transcriptomic changes in the gills highlighted the different countermeasures the salmon mounted to address the threat of the chemotherapeutics. The molecules identified both at the mucosal and systemic levels, especially those with known functions in antioxidant defence, and ROS scavenging and detoxification underscored how critical responses initiated a complex signalling in a broad variety of cellular processes. The chemical oxidant may also interfere with liver function, and thus its use must be considered with caution, though minimal risks were documented in terms of performance and ability to respond to a secondary stressor. Our results offer new mechanistic insights into salmon’s physiological responses and adaptations to chemotherapeutics-induced oxidative stress. This information will be beneficial for designing optimised treatment strategies for using oxidative biocides such as PAA in fish farming.

## Figures and Tables

**Figure 1 antioxidants-10-01931-f001:**
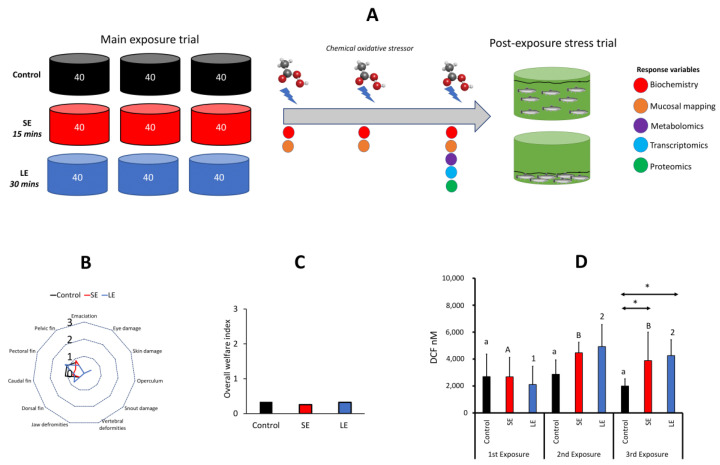
Experimental set-up, external welfare status, and plasma reactive oxygen species levels. (**A**) Fish were exposed to the oxidant every 15 days over a period of 45 days. The remaining fish were then subjected to the secondary stress of a high-density environment. Different response variables were analysed at 3 time points. (**B**,**C**) The external welfare status of the fish after 3 periodic exposures was assessed on a 0-to-3 scale system, where 0 means in good condition and 3 indicates a severely compromised state [21]. (**B**) Radial chart of the 11 indicators and (**C**) the overall welfare index based on the average score of all indicators (*n* = 9 fish per treatment group). Oxidative stress was triggered as shown in (**D**) by the level of plasma reactive oxygen species (ROS). The ROS levels were analysed 24 h after each exposure using 9 fish per treatment group. Values are presented as mean ± standard deviation. Different lower-case letters, upper case letters, and numbers indicate significant differences over time in the control, SE, and LE groups, respectively. An asterisk (*) indicates a significant difference between two groups at a particular sampling point. SE, short exposure (15 min); LE, long exposure (30 min).

**Figure 2 antioxidants-10-01931-f002:**
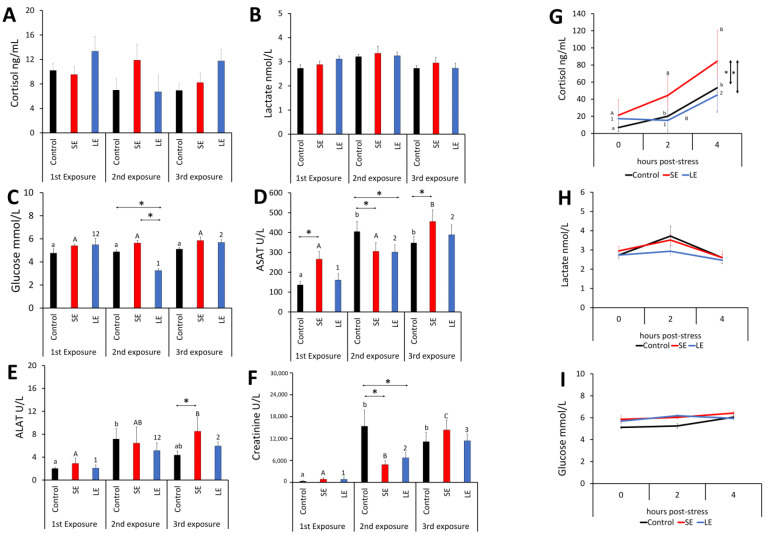
Plasma indicators for stress and organ health. The levels of (**A**) cortisol, (**B**) lactate, (**C**) glucose, (**D**) aspartate transaminase (ASAT), (**E**) alanine transaminase (ALAT), and (**F**) creatinine were measured in plasma samples (N = 9 fish per treatment group) taken 24 h after each exposure. Plasma levels of (**G**) cortisol, (**H**) lactate, and (**I**) glucose were assessed before and after the secondary stress induction. Different lower-case letters, upper case letters, and numbers indicate significant differences over time in the control, SE, and LE groups, respectively. An asterisk (*) indicates a significant difference between two groups at a particular sampling point. For the purpose of clarity, this was only indicated at 4 h post-stress in (**G**) but a similar trend of significant inter-treatment differences was likewise identified at 2 h post stress. SE, short exposure (15 min); LE, long exposure (30 min). Values are presented as mean ± standard deviation of 9 individual fish per treatment group.

**Figure 3 antioxidants-10-01931-f003:**
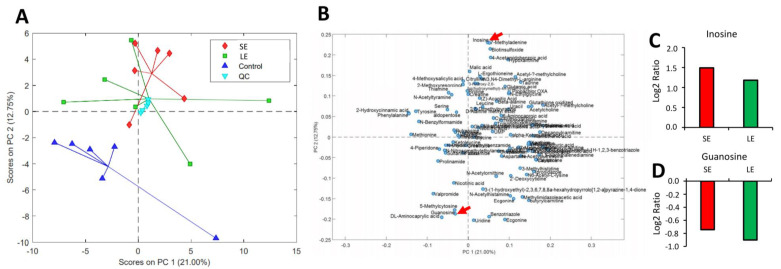
Plasma metabolomes of oxidant-exposed salmon smolts after the 3rd exposure. (**A**) Score plot from a principal component analysis (PCA) model calculated on the relative concentrations of the compounds annotated at Level 1, 2a, or 2b in the reduced dataset (see Appendix A). (**B**) Loading plot from the PCA model calculated on the relative concentrations of the compounds annotated at Level 1, 2a, or 2b in the reduced dataset. Data presented in A and B have been auto scaled. (**C**) Inosine and (**D**) guanosine were the two metabolites significantly affected in both the SE and LE groups. Values are given as the Log2 ratio relative to the control (N = 6 fish per treatment group). SE, short exposure (15 min); LE, long exposure (30 min), QC, quality control.

**Figure 4 antioxidants-10-01931-f004:**
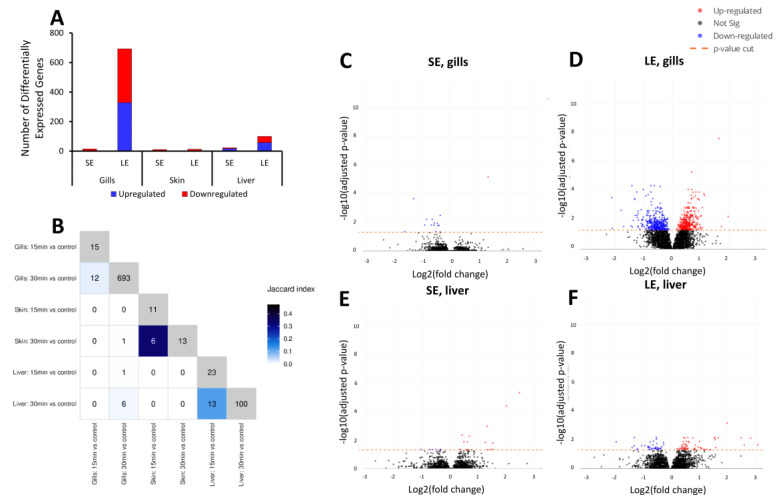
Transcriptomic changes in the gills, skin, and liver of oxidant-exposed salmon smolts. Tissues were collected after the 3rd exposure and subjected to a microarray analysis. (**A**) Differentially expressed genes, distinctively identified as either up- or down-regulated relative to the control group. (**B**) Heatmap demonstrating the overlap of different comparisons. Note that the numbers on the diagonal represent the total number of selected features found for each contrast. The colours of the squares represent the Jaccard index (the intersection over the union) for the contrasts on the *x*-axis with those on the *y*-axis. (**C**–**F**) Representative volcano plots of the (**C**,**D**) gill and (**E**,**F**) liver transcriptomes showing significance (as −log10 transformed *p*-values) against magnitude (Log2[fold change]). Features identified as having different levels between samples are represented as red (up-regulated) or blue (down-regulated) dots. SE, short exposure (15 min); LE, long exposure (30 min). Six individual fish per tissue were used.

**Figure 5 antioxidants-10-01931-f005:**
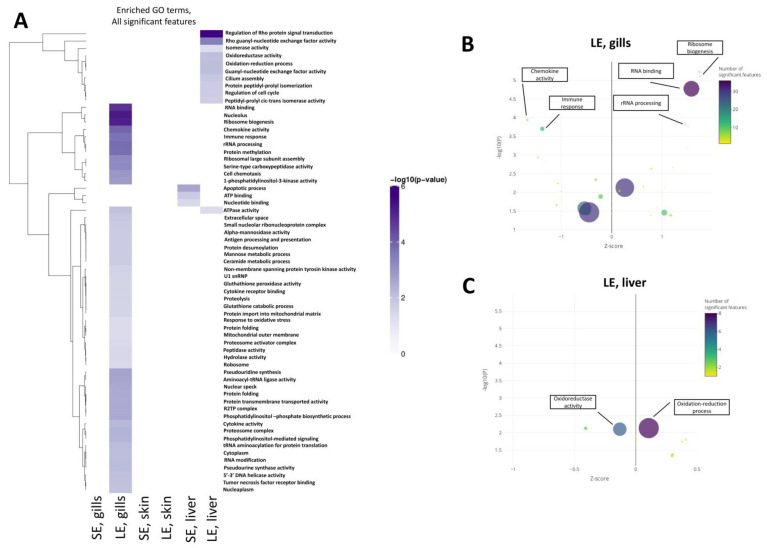
Gene ontology (GO) analysis of differentially expressed genes in the gills, skin, and liver of oxidant-exposed salmon smolts. (**A**) Heatmap of significantly enriched GO terms. Comparisons are shown on the X axis with GO terms on the Y axis. Colour is assigned based on the −log10(enrichment *p*−value), with lighter colours implying less significant enrichment. Hierarchical clustering was applied to the terms (rows). The most significant terms were clustered according to Euclidean distance using the complete linkage method. (**B**,**C**) Representative bubble plots of enriched terms showing all significant features. Only the transcriptomes of the (**B**) gills and (**C**) liver from the LE group are shown here. Enrichment analyses with the enrichment Z-score on the *x*-axis and −log10(*p*−value) on the *y*-axis. Point size represents term size, and point colour represents the calculated Z-score. Some GO terms have been highlighted. SE, short exposure (15 min); LE, long exposure (30 min).

**Figure 6 antioxidants-10-01931-f006:**
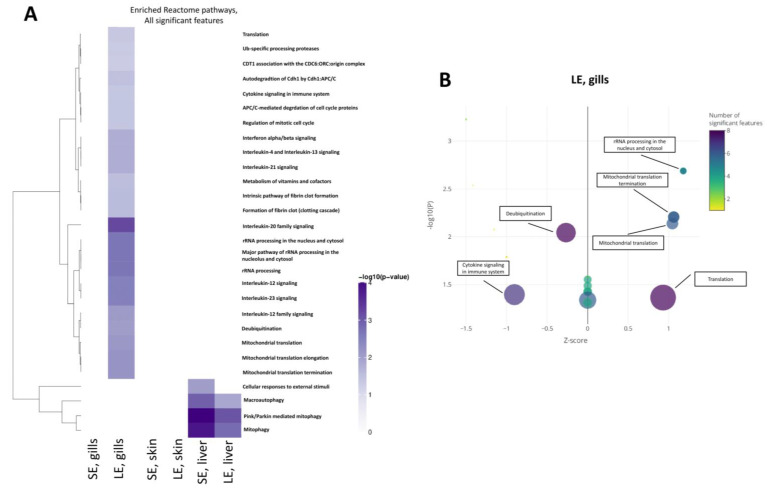
Reactome pathways of differentially expressed genes induced by oxidant exposure. (**A**) Heatmap of significantly enriched Reactome pathways. Significant features (at adjusted *p*−value < 0.05) from each contrast were analysed for an enrichment of Reactome pathway membership using a hypergeometric test by mapping features to genes (if appropriate). Enrichment (*p*-value < 0.05) was assessed for the combination of selected features. (**B**) Bubble plot of enriched pathways in the gills of fish from the LE group. SE, short exposure (15 min); LE, long exposure (30 min).

**Figure 7 antioxidants-10-01931-f007:**
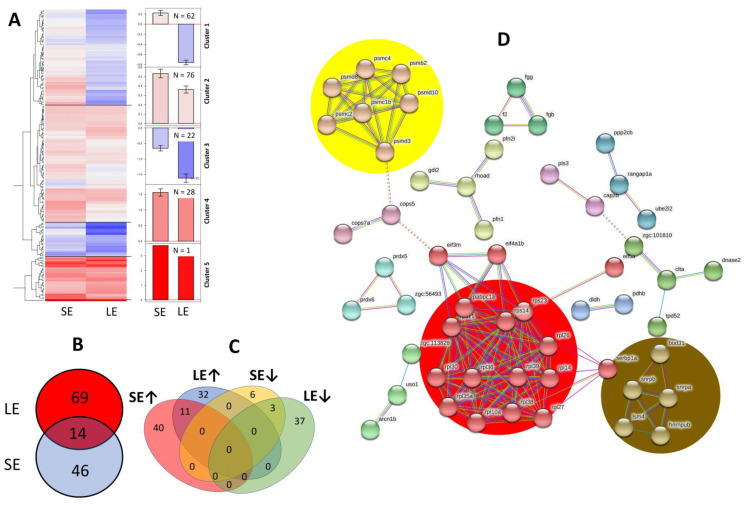
Skin mucus proteome of salmon smolts after the 3rd oxidant exposure. (**A**) Heatmap showing the relative down- and up-regulation of identified proteins (colours from blue to red) that were differentially regulated in the SE and LE groups relative to the control (N = 6 per group). The heatmap was divided into 5 sub-clusters, and the means of the respective expression values with SEM-error bars are shown as bar plots in the middle (n: number of proteins in a cluster). (**B**,**C**) Venn diagrams showing the associations of differentially expressed proteins in the different treatment groups. In (**B**), overlap did not account for the direction of change, while in (**C**), overlaps were clearly classified according to the direction of change. (**D**) Protein interaction map of identified skin mucus proteins. A possible protein-protein interaction map with high edge confidence was generated by string v.11. Joining lines represent a confidence of 0.700/1. The protein interaction network was created using zebrafish orthologs of the proteins identified in the salmon skin mucus. Accession numbers of the zebrafish proteins used in the construction of the interaction map are provided in Appendix A. SE, short exposure (15 min); LE, long exposure (30 min).

## Data Availability

The data presented in this study are available in article and Appendix A.

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
