# Peer review of "Multiomics Provide Insights into the Key Molecules and Pathways Involved in the Physiological Adaptation of Atlantic Salmon (Salmo salar) to Chemotherapeutic-Induced Oxidative Stress"

_antioxidants, 2021, doi:10.3390/antiox10121931_

Round 1
Reviewer 1 Report
The present study revealed the physiological response induced by exposure to oxidative chemotherapeutic peracetic acid through various metabolomics approach. and suggested key molecule as biomarker and physiological adaptation pathway in Atlantic salmon. The approach seemed to be novel and the study have high scientific value.
The main question addressed in this manuscript is to elucidate the physiological consequences of repeated exposures to the oxidative biocide PAA in Atlantic salmon. As mentioned in Introduction by the authors, many of studies on the effects of chemicals in fish have been conducted by single exposure. Actually, farmed fish are exposed to therapeutic agent repeatedly. The experimental design in this manuscript is practical to know the effects of therapeutics on the physiological consequences of fish. Therefore, the setting of the question and experimental design are relevant and interesting for readers in the field of aquaculture.
Conventional approach in this type of study is conducted by examine the change of histopathological and/or biochemical analysis. In this manuscript, authors are approached by various metabolomics technique. This is quite unique and authors are succeeded to provide novel findings.
other comments:
- the paper was well written, the text was clear and easy to read
Author Response
Reviewer 1
The present study revealed the physiological response induced by exposure to oxidative chemotherapeutic peracetic acid through various metabolomics approach. and suggested key molecule as biomarker and physiological adaptation pathway in Atlantic salmon. The approach seemed to be novel and the study have high scientific value.
The main question addressed in this manuscript is to elucidate the physiological consequences of repeated exposures to the oxidative biocide PAA in Atlantic salmon. As mentioned in Introduction by the authors, many of studies on the effects of chemicals in fish have been conducted by single exposure. Actually, farmed fish are exposed to therapeutic agent repeatedly. The experimental design in this manuscript is practical to know the effects of therapeutics on the physiological consequences of fish. Therefore, the setting of the question and experimental design are relevant and interesting for readers in the field of aquaculture.
Conventional approach in this type of study is conducted by examine the change of histopathological and/or biochemical analysis. In this manuscript, authors are approached by various metabolomics technique. This is quite unique and authors are succeeded to provide novel findings.
other comments:
the paper was well written, the text was clear and easy to read
Response:
We would like to thank the reviewer for the positive feedback, specifically by highlighting the scientific novelty of the present study. We hope that this manuscript will be instrumental in advancing our understanding of the physiological cost of therapeutic interventions in fish.

Reviewer 2 Report
MS ID: antioxidants-1421807
MS type: Article
MS Title: Multiomics provide insights into the key molecules and pathways involved in the physiological adaptation of Atlantic salmon (Salmo salar) to chemotherapeutic-induced oxidative stress
Reviewer Reports: This is a very interesting manuscript that investigated the key molecules and processes that enables Atlantic salmon evade and adapt to chemotherapapeutic- induced stress. The authors exposed Salmon to either short or long stress to repeated exposure periods. They found that longer exposure duration affected the fish most and with gills as the most vulnerable organ. The manuscript has potential for high impact to the field. However, I have the following major concerns that the authors need to consider
Major Concerns:
- What is the physiological relevant dose of chemical that the fish encounter in the farm or in natural condition? This need to be clearly mentioned in the manuscript.
- The authors mentioned that they decided to double the dose from 5 mg/L to 10 mg/L any reason for increasing the dose. What was the exposure duration for the 5 mg/L experiment?
- The photoperiod in this experiment was set at 24h L:0D. Is this condition normal for salmon farming and how sustainable is the condition in the farm?
- For the acute post exposure trial where the water levels were acutely depleted to induced stress: was the concentration of the chemotherapeutic stressor the same as before or elevated in this situation? Second did the authors check and monitor the oxygen concentration in the tank? These fish most likely would have experience additional stress such as lack of oxygen.
- Do the authors have explanation why the lactate levels (Figure 2H) decreased after initial increase post stress level?
- The authors mentioned in the discussion that fish can regulate their internal ROS homeostasis but did not state how fish regulate their ROS please explain what mechanisms did the fish use to ensure ROS homeostasis.
- What is the half-life of Cortisol in fish?- another possible explanation could be that in the longer duration effect of cortisol has cleared so could not be capture?
Author Response
Reviewer 2
MS Title: Multiomics provide insights into the key molecules and pathways involved in the physiological adaptation of Atlantic salmon (Salmo salar) to chemotherapeutic-induced oxidative stress
Reviewer Reports: This is a very interesting manuscript that investigated the key molecules and processes that enables Atlantic salmon evade and adapt to chemotherapapeutic- induced stress. The authors exposed Salmon to either short or long stress to repeated exposure periods. They found that longer exposure duration affected the fish most and with gills as the most vulnerable organ. The manuscript has potential for high impact to the field. However, I have the following major concerns that the authors need to consider
Response:
We appreciate the overall positive feedback from the referee. We carefully studied the review and our response to each point raised is provided below. Changes in the manuscript are likewise highlighted by Track Changes option.
Major Concerns:
What is the physiological relevant dose of chemical that the fish encounter in the farm or in natural condition? This need to be clearly mentioned in the manuscript.
Response:
PAA is not yet being used in salmon farms as routine chemotherapeutics. At the moment, its application in salmon farming in Norway is limited to as surface disinfectants. In the last years, we have been working on identifying the consequences of PAA application in the hope of establishing its an evidence-use as a chemotherapeutics. We have reported earlier that salmon smolts have been exposed to concentration 0.6 to 4.8 ppm (https://www.sciencedirect.com/science/article/pii/S1050464819308484, https://www.sciencedirect.com/science/article/pii/S0044848620317932). In addition, 8.9 ppm was tested in another salmonid, rainbow trout (https://www.mdpi.com/2410-3888/3/1/10). PAA have been tested in various fish species from 0-14 ppm, either by single exposure or continuous exposure (https://onlinelibrary.wiley.com/doi/abs/10.1111/jwas.12475). These publications are included in the paper and a statement in the introduction has been added to reflect this.
The authors mentioned that they decided to double the dose from 5 mg/L to 10 mg/L any reason for increasing the dose. What was the exposure duration for the 5 mg/L experiment?
Response:
We doubled the tested concentration from the previously reported concentration (https://www.sciencedirect.com/science/article/pii/S0044848620317932) to fully document the extent of PAA effects on salmon physiology. As reported in Lazado et al. 2021, salmon smolts were exposed to 4.8 ppm PAA for 30 mins, but this was only a single exposure event.
The photoperiod in this experiment was set at 24h L:0D. Is this condition normal for salmon farming and how sustainable is the condition in the farm?
Response:
Continuous lighting is the normal photoperiod condition in tank-based rearing of salmon, especially after they have smoltified. This photoperiod regime is often used to delay sexual maturation of salmon reared in tanks. Once they are transferred to sea cages, the photoperiod follows the natural photoperiod of the location.
For the acute post exposure trial where the water levels were acutely depleted to induced stress: was the concentration of the chemotherapeutic stressor the same as before or elevated in this situation? Second did the authors check and monitor the oxygen concentration in the tank? These fish most likely would have experience additional stress such as lack of oxygen.
Response:
The fish were not simultaneously exposed to the chemical stressor and the density stress. We believe that we have clearly described this in the methodology. The stress experiment was performed after the last PAA exposure event. The oxygen was checked while stress experiment was on-going and ensured that it was above 90% saturation. This information was added in the revised test.
Do the authors have explanation why the lactate levels (Figure 2H) decreased after initial increase post stress level?
Response:
The changes in lactate level in Figure 2H were not statistically significant. Even though there was a visual indication of change, we could not draw physiological implications conclusively since no significant difference was detected.
The authors mentioned in the discussion that fish can regulate their internal ROS homeostasis but did not state how fish regulate their ROS please explain what mechanisms did the fish use to ensure ROS homeostasis.
Response:
In the discussion, we discussed that the alterations in the transcriptomes and metabolomes, especially in the molecules with known involvement in antioxidative response and radical detoxification, the mechanisms the fish mount to counteract the ROS imbalance due to chemotherapeutics-induced oxidative stress. We added a statement in the conclusion to further highlight this, as the reviewer emphasised. Likewise, we made some changes in some parts of the discussion to further emphasise this.
What is the half-life of Cortisol in fish?- another possible explanation could be that in the longer duration effect of cortisol has cleared so could not be capture?
Response:
Salmon plasma cortisol usually peaked during the first 6 hours after stress induction, but this is dependent on the type of stressor. The cortisol data presented in Figure 2A were the cortisol levels 24 h after stress. It could be possible that the responses from both SE and LE had already returned to the basal level. Cortisol response in Figure 2G clearly showed the elevation of cortisol for both SE and LE indicating that the groups, despite having been exposed to an oxidant periodically, were still able to mount a classical stress response.

Reviewer 3 Report
No further comments or suggestions to the authors
Author Response
Reviewer 3
No further comments or suggestions to the authors
Response
We acknowledge the positive feedback of the reviewer on our manuscript.

Round 2
Reviewer 2 Report
The authors have addressed all my concerns.